 eLife

# Proteasome dysfunction triggers activation of SKN-1A/Nrf1 by the aspartic protease DDI-1

Nicolas J Lehrbach[1,2], Gary Ruvkun[1,2]*

[1]Department of Molecular Biology, Massachusetts General Hospital, Boston, United States; [2]Department of Genetics, Harvard Medical School, Boston, United States

**Abstract** Proteasomes are essential for protein homeostasis in eukaryotes. To preserve cellular function, transcription of proteasome subunit genes is induced in response to proteasome dysfunction caused by pathogen attacks or proteasome inhibitor drugs. In *Caenorhabditis elegans*, this response requires SKN-1, a transcription factor related to mammalian Nrf1/2. Here, we use comprehensive genetic analyses to identify the pathway required for *C. elegans* to detect proteasome dysfunction and activate SKN-1. Genes required for SKN-1 activation encode regulators of ER traffic, a peptide N-glycanase, and DDI-1, a conserved aspartic protease. DDI-1 expression is induced by proteasome dysfunction, and we show that DDI-1 is required to cleave and activate an ER-associated isoform of SKN-1. Mammalian Nrf1 is also ER-associated and subject to proteolytic cleavage, suggesting a conserved mechanism of proteasome surveillance. Targeting mammalian DDI1 protease could mitigate effects of proteasome dysfunction in aging and protein aggregation disorders, or increase effectiveness of proteasome inhibitor cancer chemotherapies.

*For correspondence: ruvkun@ molbio.mgh.harvard.edu

**Competing interests:** The authors declare that no competing interests exist.

## Introduction

The proteasome is a multi-protein complex responsible for the majority of protein degradation in eukaryotic cells (*Tomko and Hochstrasser, 2013*). The essential function of the proteasome, and its highly conserved structure and mechanism of proteolysis renders it an attractive target for bacteria and other competitors. Production of small molecule inhibitors and protein virulence factors that target the proteasome by some bacteria and fungi exploits this vulnerability to gain a growth advantage (*Fenteany et al., 1995*; *Groll et al., 2008*; *Meng et al., 1999*). In addition, environmental stresses antagonize the proteasome by causing accumulation of unfolded and aggregated proteins that can form a non-productive inhibitory interaction with proteasomes (*Ayyadevara et al., 2015*; *Deriziotis et al., 2011*; *Kristiansen et al., 2007*; *Snyder et al., 2003*). Human diseases in which proteasome dysfunction is implicated highlight the importance of maintaining proteasome function in the face of these challenges (*Ciechanover and Kwon, 2015*; *Paul, 2008*; *Tomko and Hochstrasser, 2013*), and it follows that animal cells possess mechanisms to monitor and defend proteasome function.

A conserved response to proteasome disruption is the transcriptional up-regulation of proteasome subunit genes (*Fleming, 2002*; *Meiners et al., 2003*; *Wójcik and DeMartino, 2002*). In mammalian cells members of the Cap' n' Collar basic leucine zipper (CnC-bZip) family of stress responsive transcription factors mediate this transcriptional response. Two CnC-bZip transcription factors, Nrf1/NFE2L1 and Nrf2, have similar DNA-binding domains and may regulate an overlapping set of downstream targets. However, only Nrf1 is required for upregulation of proteasome subunits following proteasome disruption, whereas Nrf2 may activate proteasome expression under other circumstances (*Arlt et al., 2009*; *Radhakrishnan et al., 2010*; *Steffen et al., 2010*). The events leading

**eLife digest** Proteins perform many important roles in cells, but these molecules can become toxic if they are damaged or are no longer needed. A molecular machine called the proteasome destroys 'unwanted' proteins in animal and other eukaryotic cells. If the proteasome stops working properly, unwanted proteins start to accumulate and cells respond by increasing the activity of genes that make proteasomes. A protein called SKN-1 is involved in this response and activates the genes that encode proteasome proteins, but it is not understood how SKN-1 "senses" that proteasomes are not working properly.

Here, Lehrbach and Ruvkun used a roundworm called *Caenorhabditis elegans* to search for new genes that activate SKN-1 when the proteasome's activity is impaired. The roundworms were genetically engineered to produce a fluorescent protein that indicates when a particular gene needed to make proteasomes is active. Lehrbach and Ruvkun identified some roundworms with mutations that cause the levels of fluorescence to be lower, indicating that SKN-1 was less active in these animals. Further experiments showed that some of these mutations are in genes that encode enzymes called DDI-1 and PNG-1. DDI-1 is able to cut certain proteins, while PNG-1 can remove sugars that are attached to proteins. Therefore, it is likely that these enzymes directly interact with SKN-1 and alter it to activate the genes that produce the proteasome.

More work is now needed to understand the details of how modifying SKN-1 changes its activity in cells. In the future, drugs that target DDI-1 or PNG-1 might be used to treat diseases in which proteasome activity is too high or low, including certain cancers and neurodegenerative diseases.

to Nrf1 activation in response to proteasome disruption are complex. In vitro analyses in human and mouse cells indicate that Nrf1 is an endoplasmic reticulum (ER) membrane associated glycoprotein that is constitutively targeted for proteasomal degradation by the ER-associated degradation (ERAD) pathway. Upon proteasome inhibition Nrf1 is stabilized, undergoes deglycosylation and proteolytic cleavage, and localizes to the nucleus (*Radhakrishnan et al., 2014*; *Sha and Goldberg, 2014*; *Wang, 2006*; *Zhang and Hayes, 2013*; *Zhang et al., 2015*, *2007*, *2014*). How processing of Nrf1 is orchestrated, and its significance in responses to proteasome disruption *in vivo* are not understood.

Upon proteasome disruption, *C. elegans* induces transcription of proteasome subunit, detoxification, and immune response genes, and animals alter their behavior to avoid their bacterial food source (*Li et al., 2011*; *Melo and Ruvkun, 2012*). The transcriptional response to proteasome disruption involves *skn-1*, which encodes multiple isoforms of a transcription factor with similarities to both Nrf1 and Nrf2 (*Blackwell et al., 2015*; *Li et al., 2011*). *skn-1* was originally identified for its essential role in embryonic development (*Bowerman et al., 1992*), but is also required after these early stages for stress responses in a manner analogous to mammalian Nrf1/2 (*An and Blackwell, 2003*; *Oliveira et al., 2009*; *Paek et al., 2012*). SKN-1 binds to the promoters of proteasome subunit genes and mediates their upregulation in response to proteasome disruption, and is required for survival of a mutant with attenuated proteasome function (*Keith et al., 2016*; *Li et al., 2011*; *Niu et al., 2011*). The molecular mechanism that links SKN-1 activation to the detection of proteasome dysfunction has not been established.

Here, we use genetic analysis to uncover the mechanism that couples detection of proteasome defects to these transcriptional responses in *C. elegans*. We find that an ER-associated isoform of SKN-1 (SKN-1A), is essential for this response. Our genetic data show that the ER-association of this transcription factor normally targets it for poteasomal degradation via ERAD, but is also required for its correct post-translational processing and activation during proteasome dysfunction. After ER-trafficking, our data argues that the PNG-1 peptide N-glycanase removes glycosylation modifications that occur in the ER, and then the DDI-1 aspartic protease cleaves SKN-1A. Each of these steps in SKN-1A processing is essential for the normal response to proteasomal dysfunction. This pathway is essential for compensation of proteasome function under conditions that partially disrupt the proteasome; when compensation is disabled, mild inhibition of the proteasome causes lethal arrest of

development. Thus we reveal a vital mechanism of proteasome surveillance and homeostasis in animals.

## Results

### The aspartic protease DDI-1 and ERAD factors are required for transcriptional responses to proteasome disruption

The proteasome subunit gene *rpt-3* is upregulated in a *skn-1*-dependent manner in response to proteasome disruption (*Li et al., 2011*). We generated a chromosomally integrated transcriptional reporter in which the *rpt-3* promoter drives expression of GFP (*rpt-3::gfp*). This reporter gene is upregulated in response to drugs such as bortezomib or mutations that cause proteasome dysfunction. To identify the genetic pathways that sense proteasome dysfunction and trigger the activation of SKN-1, we took advantage of a regulatory allele affecting the *pbs-5* locus. *pbs-5* encodes the *C. elegans* ortholog of the beta 5 subunit of the 20S proteasome. The *pbs-5(mg502)* mutation causes constitutive *skn-1*-dependent activation of *rpt-3::gfp* expression, but does not otherwise alter fertility or viability (*Figure 1—figure supplement 1*). Following EMS mutagenesis, we isolated a collection of recessive mutations that suppress the activation of *rpt-3::gfp* caused by *pbs-5(mg502)*, and identified the causative mutations by whole genome sequencing (*Table 1*). The collection includes multiple alleles of genes encoding factors required for ERAD. In ERAD, misfolded glycoproteins are retrotranslocated from the ER lumen to the cytoplasm, where they are degraded by the proteasome (*Smith et al., 2011*). We isolated 3 alleles of *sel-1*, a gene that encodes the *C. elegans* orthologue of HRD3/SEL1. HRD3/SEL1 localizes to the ER membrane and recognizes ERAD substrates in the ER (*Carvalho et al., 2006*; *Denic et al., 2006*; *Gauss et al., 2006*), and a single allele of *sel-9*, which encodes the *C. elegans* orthologue of TMED2/EMP24, which is also ER-localized and implicated in ER quality control (*Copic et al., 2009*; *Wen and Greenwald, 1999*). We also found mutations in *png-1*, which encodes the *C. elegans* orthologue of PNG1/NGLY1. After ERAD substrates have been retrotranslocated to the cytoplasm, PNG1/NGLY1 removes N-linked glycans to allow their degradation by the proteasome (*Kim et al., 2006*; *Suzuki et al., 2016*). Most strikingly, we isolated six alleles of *C01G5.6* (hereafter *ddi-1*), which encodes the *C. elegans* orthologue of DDI1 (DNA damage inducible 1). DDI-1 is an aspartic protease, highly conserved throughout eukaryotes (*Sirkis et al., 2006*). DD1's function is poorly understood, but it has been implicated in regulation of proteasome function and protein secretion (*Kaplun et al., 2005*; *White et al., 2011*).

We examined activation of *rpt-3::gfp* in ERAD and *ddi-1* mutant animals following disruption of proteasome function by RNAi of the essential proteasome subunit *rpt-5*. *rpt-5(RNAi)* caused larval arrest confirming that all genotypes are similarly susceptible to RNAi. While *rpt-5(RNAi)* causes

**Table 1.** EMS-induced mutations that disrupt *rpt-3::gfp* activation.

| Allele | Affected gene | Effect | Homologues |
|--------|---------------|--------|------------|
| mg563 | C01G5.6 | L245F | DDI1. Aspartic protease. |
| mg555 | C01G5.6 | C277S | |
| mg544 | C01G5.6 | G293R | |
| mg542 | C01G5.6 | R350STOP | |
| mg543 | C01G5.6 | M244I | |
| mg557 | C01G5.6 | L334F | |
| mg565 | sel-1 | G594E | HRD3/SEL1. ER membrane protein, required for ERAD substrate recognition. |
| mg567 | sel-1 | A522T | |
| mg547 | sel-1 | splice site | |
| mg550 | sel-9 | S140F | EMP24/TMED2. ER membrane protein. |
| mg561 | png-1 | G498R | PNG1/NJLY1. Peptide N-glycanase. Removes N-linked glycans during ERAD. |
| mg564 | png-1 | splice site | |

robust activation of *rpt-3::gfp* in wild-type animals, mutants lacking ERAD factors or *ddi-1* failed to fully activate *rpt-3::gfp* (*Figure 1a*). The requirement for *sel-11*, which encodes an ER-resident ubiquitin ligase required for ERAD (*Smith et al., 2011*), supports a general requirement for ERAD in activation of *rpt-3::gfp* expression. These genes are also required for upregulation of *rpt-3::gfp* following proteasome disruption by bortezomib (data not shown).

Unlike the wild type, mutants defective in ERAD, or lacking DDI-1, arrested or delayed larval development in the presence of low doses of bortezomib (*Figure 1b*). We analyzed independently derived alleles of *png-1* and *ddi-1*, indicating that hypersensitivity to proteasome inhibition is unlikely to be a consequence of linked background mutations. *png-1* animals consistently showed

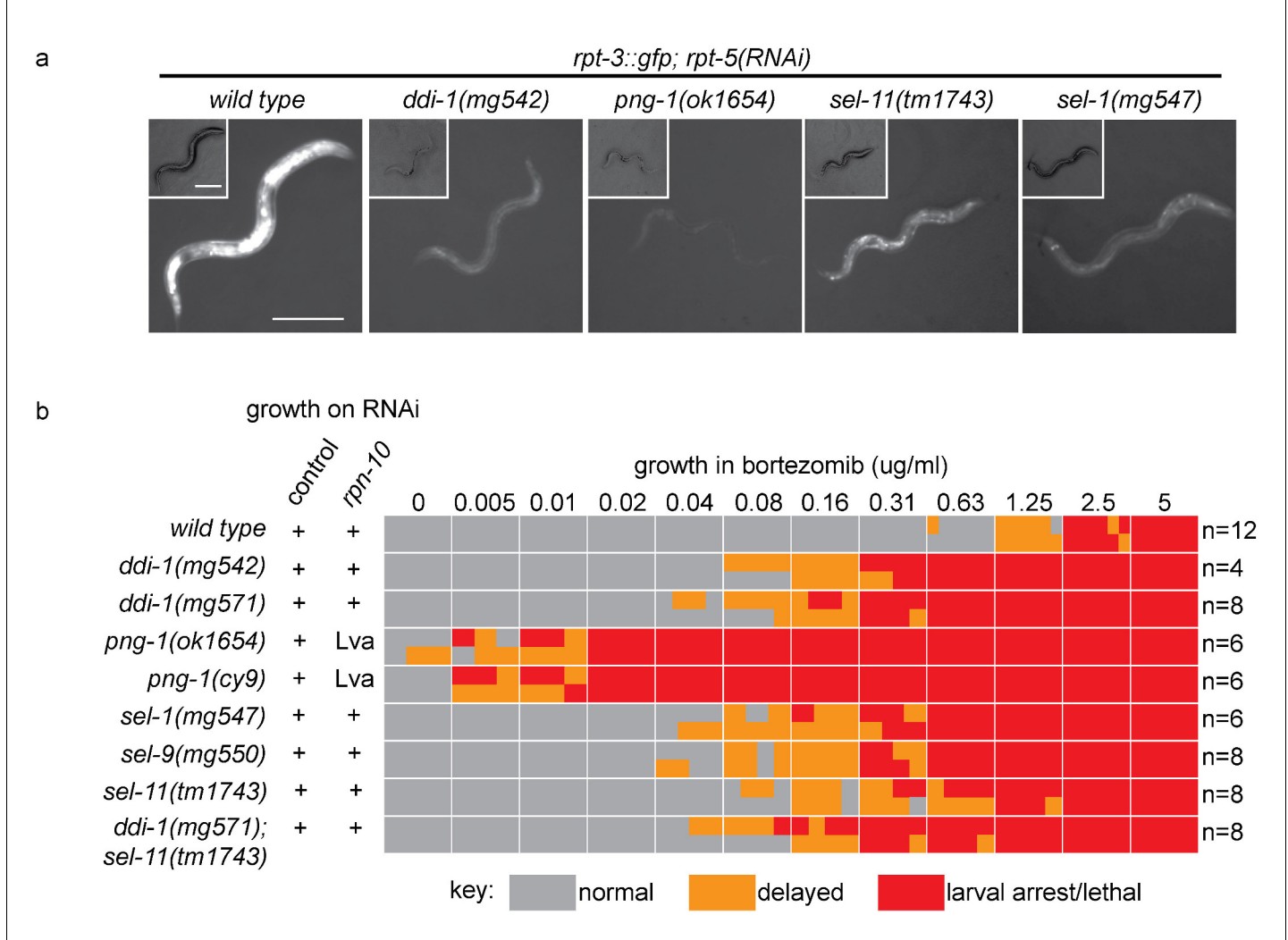

**Figure 1.** ER-associated degradation factors and the aspartic protease DDI-1 are required for responses to proteasome disruption. (**a**) *rpt-3::gfp* expression following disruption of proteasome function by *rpt-5(RNAi)* in various mutant backgrounds. Scale bars 100 μm. (**b**) Table showing growth vs. arrest phenotypes of various mutants in the presence of bortezomib or upon *rpn-10* RNAi. For RNAi experiments, L1 animals were incubated for 3 days on indicated RNAi plates, and scored for developmental arrest (+; normal development, Lva; larval arrest). For bortezomib experiments, ~15 L1 animals were incubated for 4 days in liquid cultures containing varying concentrations of bortezomib, and scored for developmental progression. The number (n) of replicate bortezomib experiments performed for each genotype is shown on the right. Each colored rectangle is divided into equal parts to show results from each replicate.

The following figure supplement is available for figure 1:

**Figure supplement 1.** *skn-1*-dependent activation of *rpt-3::gfp* in *pbs-5(mg502)* mutants.

the most severe defect, and were unable to grow in the presence of very low concentrations of bortezomib. Consistent with their drug sensitivity, mild disruption of proteasome function by RNAi-mediated depletion of the non-essential proteasome subunit RPN-10 causes a synthetic larval lethal phenotype in animals mutant for *png-1*. The bortezomib sensitivity of *ddi-1; sel-11* double mutants was not enhanced compared to that of *ddi-1* single mutants, suggesting that *ddi-1* and ERAD factors act in the same genetic pathway. We conclude that ERAD and DDI-1 are required for transcriptional upregulation of proteasome subunits and survival during proteasome dysfunction. Given the defective activation of *rpt-3::gfp*, a direct target of SKN-1, it is likely that upon proteasome disruption, these factors are required to activate SKN-1.

## SKN-1A, an ER-associated isoform of SKN-1 mediates proteasome homeostasis

The *skn-1* gene generates 3 protein isoforms using alternative transcription start sites (SKN-1A, SKN-1B, and SKN-1C; *Figure 2a*). The isoforms share an identical C-terminal DNA binding domain, but differ in their N-termini. *skn-1(RNAi)* targets sequences common to all three transcripts. Both *skn-1(RNAi)*, and the *skn-1(zu67)* mutation cause defective responses to proteasome dysfunction (NL unpublished). *skn-1(zu67)* is a nonsense allele affecting an exon shared by SKN-1A and SKN-1C, but that does not affect SKN-1B, suggesting SKN-1A and/or SKN-1C are required. SKN-1C encodes a 61 kD protein expressed specifically in the intestine, and SKN-1A encodes a 71 kD protein expressed in most tissues (*An and Blackwell, 2003*; *Bishop and Guarente, 2007*; *Staab et al., 2014*). SKN-1A differs from SKN-1C solely by the presence of 90 additional amino acids at the N-terminus that includes a predicted transmembrane domain (*Figure 2b*), and SKN-1A has been found to associate with the ER (*Glover-Cutter et al., 2013*).

We used CRISPR/Cas9 to generate an isoform-specific genetic disruption of SKN-1A, by introducing premature stop codons to the *skn-1a* specific exons of the *skn-1* locus (hereafter referred to as *skn-1a* mutants). Homozygous *skn-1a* mutant animals are viable, and under standard conditions show a growth rate and fertility indistinguishable from the wild type. However, *skn-1a* mutant animals fail to activate rpt-3::gfp in the *pbs-5(mg502)* mutant background, or upon RNAi of essential proteasome subunit genes, or exposure to bortezomib (*Figure 2c,d*, data not shown). We note that in these experiments *skn-1a* mutants failed to activate *rpt-3::gfp* in all tissues, including the intestine, where SKN-1C is expressed. Consistent with the failure to upregulate *rpt-3::gfp*, *skn-1a* mutants show larval lethality when proteasome dysfunction is induced by *rpn-10(RNAi)* or treatment with a low dose of bortezomib (*Figure 2e,h*). These *skn-1a* mutations specifically affect SKN-1A, but leave SKN-1B and SKN-1C unaltered, indicating that SKN-1A is essential for normal responses to proteasome disruption and in the absence of SKN-1A, the other isoforms are not sufficient. A number of stimuli that trigger stabilization and nuclear accumulation of a transgenic SKN-1C::GFP fusion protein are known, but relatively little is known about whether these stimuli also affect SKN-1A (*Blackwell et al., 2015*). We used miniMos transgenesis (*Frøkjær-Jensen et al., 2014*) to generate genomically integrated single-copy transgenes that expresses C-terminally GFP-tagged full length SKN-1A (SKN-1A::GFP), and a second C-terminally GFP tagged truncated SKN-1A that lacks the DNA binding domain (SKN-1A[ΔDBD]::GFP). When driven by the ubiquitously active *rpl-28* promoter, we did not observe accumulation of SKN-1A::GFP or SKN-1A[ΔDBD]::GFP, consistent with constitutive degradation of these fusion proteins. Upon disruption of proteasome function, we observed stabilization and nuclear localization of SKN-1A::GFP and SKN-1A[ΔDBD]::GFP in many tissues (*Figure 2f*, *Figure 2—figure supplement 1a*). For unknown reasons, the SKN-1A[ΔDBD]::GFP transgene accumulated to higher levels than the full length transgene (*Figure 2—figure supplement 1b*). We generated similar transgenes to express tagged full length and truncated SKN-1C, but did not observe any effect of proteasome disruption (data not shown). These data suggest proteasome dysfunction triggers activation of SKN-1A, but not SKN-1C.

We introduced the SKN-1A::GFP transgene into the *skn-1a(mg570)* and *skn-1(zu67)* mutant backgrounds. SKN-1A::GFP rescued the maternal effect lethal phenotype of *skn-1(zu67)*. SKN-1A::GFP also restored wild-type resistance to proteasome disruption, as assayed by growth on *rpn-10(RNAi)* (*Figure 3*), or growth in the presence of low concentrations of bortezomib (data not shown). This indicates that the SKN-1A::GFP fusion protein is functional, and that SKN-1A::GFP is sufficient for normal responses to proteasome dysfunction even in the absence of SKN-1C (which is disrupted by

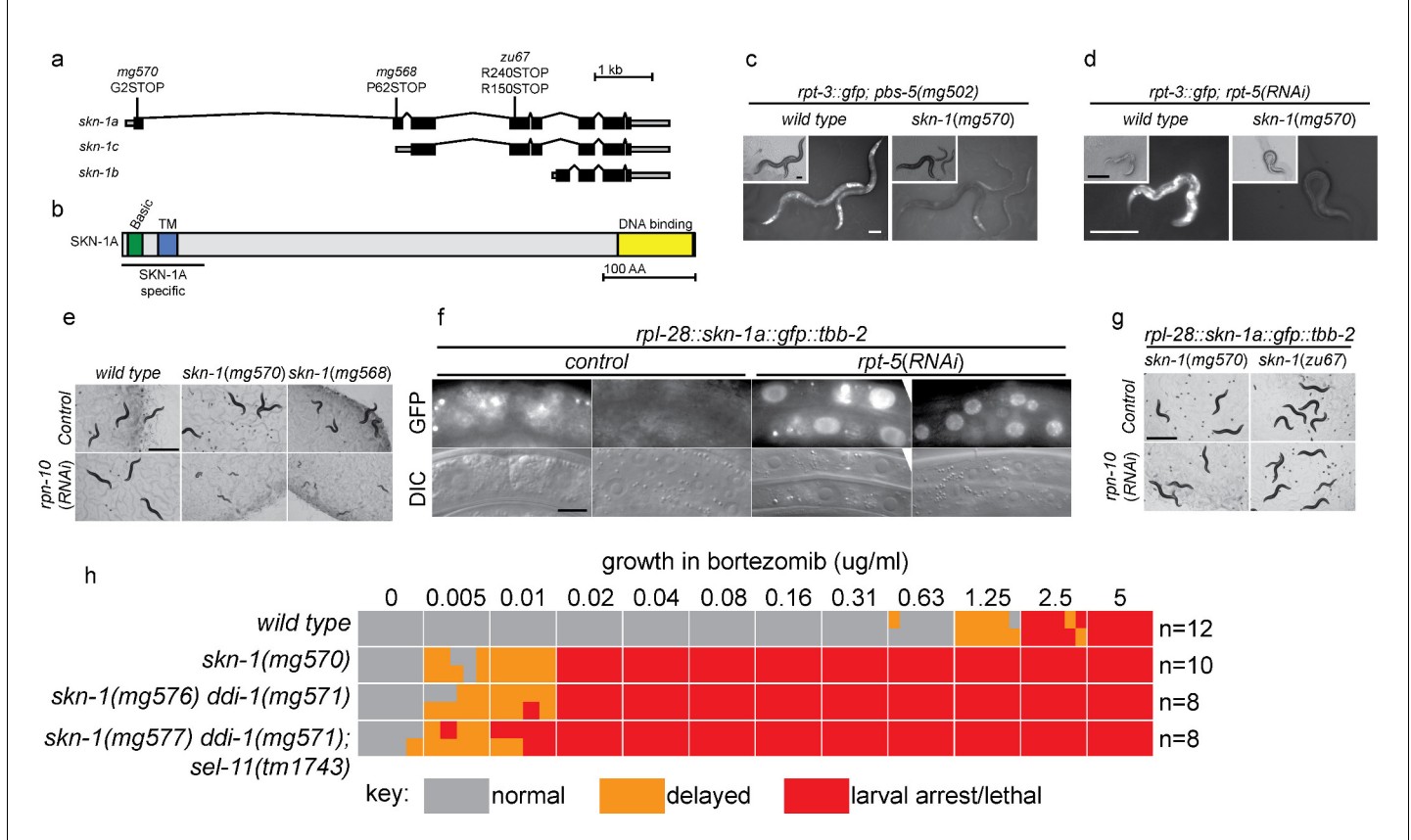

**Figure 2.** SKN-1A, a transmembrane-domain-containing isoform of SKN-1 mediates transcriptional responses to proteasome disruption. (a,b) Schematic of the (a) *skn-1* locus and (b) SKN-1A protein. In (a), the CRISPR-induced *skn-1a*-specific mutations are indicated. (c,d) *rpt-3::gfp* induction in wild type and isoform-specific *skn-1* mutants in (c) the *pbs-5(mg502)* mutant, or (d) *rpt-5(RNAi)*. Scale bars 100 μm. (e) Developmental arrest of isoform-specific *skn-1* mutants exposed to mild proteasome disruption by *rpn-10(RNAi)* but not on control RNAi. Scale bar 1 mm. (f) Expression and localization of functional SKN-1A::GFP fusion protein after proteasome disruption by *rpt-5(RNAi)*. Apparent GFP signal in control treated animals is background auto-fluorescence. Scale bar 10 μm. (g) No developmental arrest of *skn-1* mutants carrying an isoform-specific *skn-1a::gfp* transgene, and exposed to mild proteasome disruption by *rpn-10(RNAi)*. Scale bar 1 mm. (h) Table showing growth vs. arrest phenotypes of *skn-1a* mutants in in the presence of bortezomib. All *skn-1a* alleles are identical in their effect on *skn-1a* coding sequence (G2STOP). Experiments performed identically to those shown in *Figure 1b*, and data for the wild type from *Figure 1* are shown for reference.

The following figure supplement is available for figure 2:

**Figure supplement 1.** SKN-1A[ΔDBD]::GFP is stabilized upon proteasome disruption.

the *zu67* allele). As such, the transmembrane-domain-bearing SKN-1A isoform is necessary and sufficient for responses to proteasome dysfunction.

Mutation of *ddi-1* does not enhance the sensitivity of *skn-1a* mutants to bortezomib, suggesting that DDI-1 acts through SKN-1A to promote resistance to proteasome inhibitors (*Figure 2h*). Additionally removing SEL-11 weakly enhanced the bortezomib sensitivity of *ddi-1 skn-1a* double mutants, and also caused occasional growth defects even in the absence of proteasome disruption, suggesting that ERAD promotes resistance to proteasome inhibitors largely, but not solely, through regulation of SKN-1A. We examined how ERAD factors regulate SKN-1A using the SKN-1A::GFP transgenes. *sel-1* and *sel-11* mutants accumulate high levels of SKN-1A[ΔDBD]::GFP even in the absence of proteasome inhibitors, showing that SKN-1A is constitutively targeted for proteasomal degradation via ERAD (*Figure 3a*). Upon proteasome disruption, *sel-1* and *sel-11* mutants show defects in SKN-1A::GFP nuclear localization consistent with defective release from the ER (*Figure 3b*). Following proteasome disruption in *png-1* mutants SKN-1A::GFP localizes to the

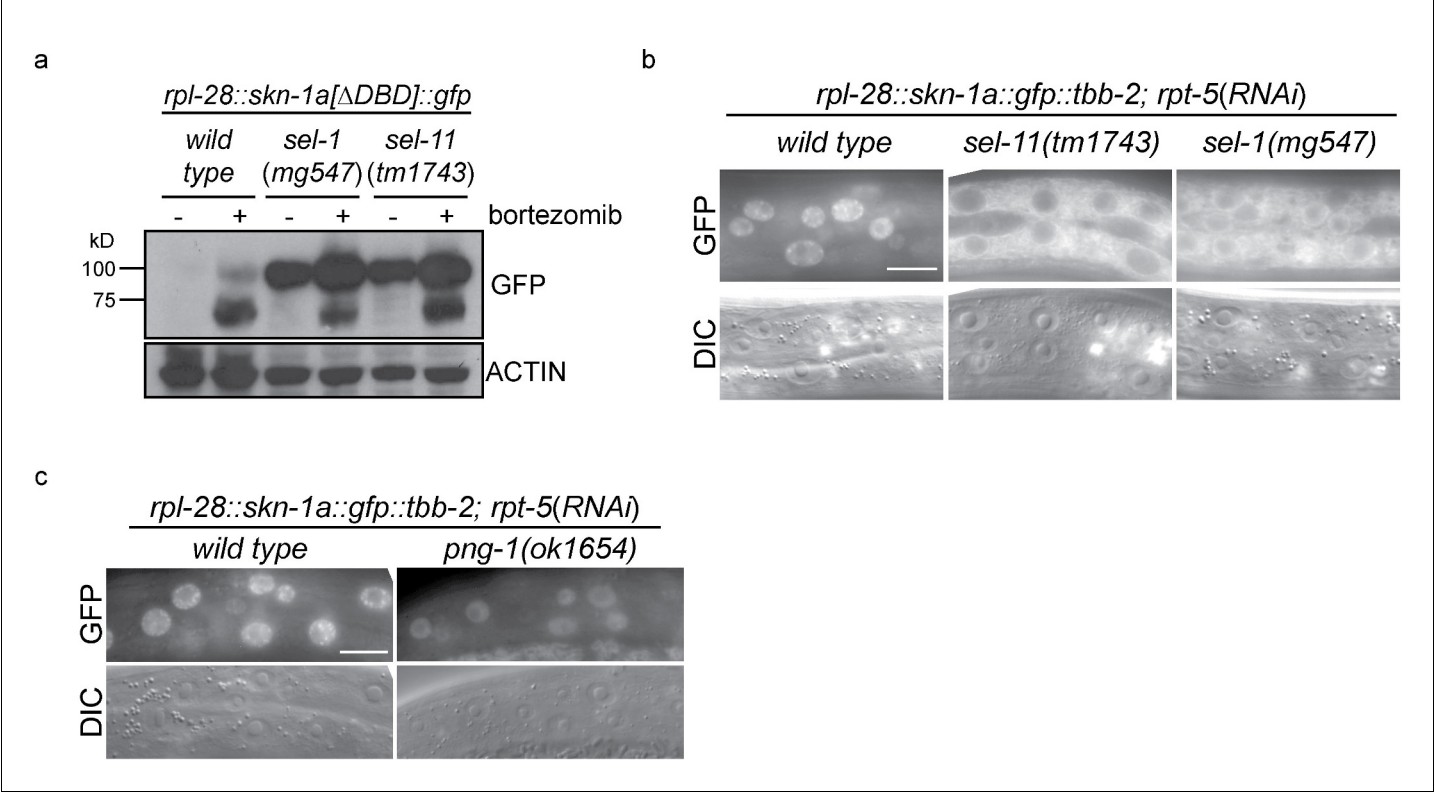

**Figure 3.** ERAD is required for constitutive SKN-1A degradation, and for activation of SKN-1A upon proteasome disruption. (a) Western blot showing expression and post-translational processing of SKN-1A[ΔDBD]::GFP in ERAD mutant animals, treated with either solvent control (DMSO) or 5 ug/ml bortezomib. SKN-1A[ΔDBD]::GFP is only detected in wild-type animals upon bortezomib exposure; a major band at ~70 kD and a minor band at ~90 kD are detected. In ERAD defective mutants, the ~90 kD band is strongly detected under all conditions, and the ~70 kD band appears only following bortezomib treatment. Actin is used as a loading control. (b) Expression and localization of SKN-1A::GFP in wild type and *sel-1* and *sel-11* ERAD defective mutants after proteasome disruption by *rpt-5(RNAi)*. In ERAD defective mutants, SKN-1A::GFP fails to localize to the nucleus. Scale bar 10 μm. (c) Expression and localization of SKN-1A::GFP in wild type and *png-1* mutants after proteasome disruption by *rpt-5(RNAi)*. In *png-1* mutants, SKN-1A:: GFP is able to localize to the nucleus, although at reduced levels compared to the wild-type. Scale bar 10 μm.

nucleus, indicating PNG-1 acts downstream of release from the ER (*Figure 3b*). Lower levels of SKN-1A::GFP accumulate in the nuclei of *png-1* mutants than in the wild type, but this mild effect is unlikely to fully account for the severely defective responses to proteasome inhibition in *png-1* mutant animals, suggesting retention of glycosylation modifications normally removed by PNG-1 likely disrupts SKN-1A's nuclear function. These data suggest that activation of ER-associated and N-glycosylated SKN-1A is required for responses to proteasome dysfunction.

## DDI-1 aspartic protease localizes to both nucleus and cytoplasm, and is upregulated upon proteasome disruption

To examine the expression and subcellular localization of the DDI-1 protease, we used miniMos to generate a single copy integrated transgene expressing full length DDI-1 fused to GFP at the N-terminus, under the control of the *ddi-1* promoter. The GFP::DDI-1 fusion protein is expressed in most tissues and shows diffuse cytoplasmic and nuclear localization under control conditions, and can rescue a *ddi-1* mutant (see below). Following disruption of proteasome function by *rpt-5(RNAi)*, GFP:: DDI-1 expression is dramatically induced, and GFP::DDI-1 is enriched in nuclei (*Figure 4a*). We used CRISPR/Cas9 to modify the *ddi-1* locus to incorporate an HA epitope tag near the N-terminus of endogenous DDI-1. Following bortezomib treatment of *ddi-1(mg573[HA::ddi-1])* animals, we observed strong upregulation (greater than 10-fold, based on blotting of diluted samples) of the HA-tagged endogenous DDI-1 (*Figure 4b*). The *ddi-1* promoter contains a SKN-1 binding site (*Niu et al., 2011*). Upregulation of GFP::DDI-1 by *rpt-5(RNAi)* is greatly reduced in *skn-1a(mg570)*

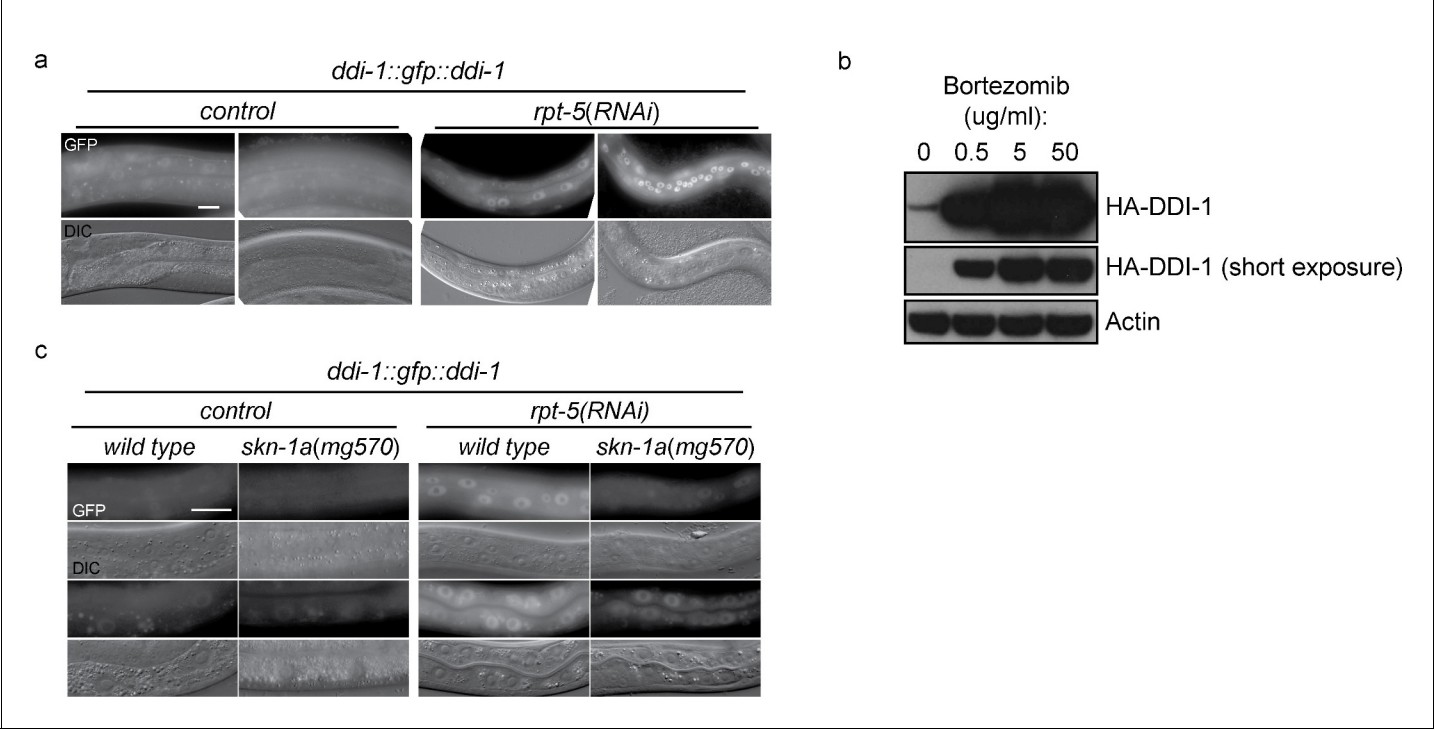

**Figure 4.** DD1-1 is upregulated upon proteasome disruption. (**a**) A functional GFP::DDI-1 fusion protein is strongly induced and localizes to the nucleus upon proteasome disruption by *rpt-5(RNAi)*. Scale bar 20 µm. (**b**) Western blot showing induction of HA-tagged endogenous DDI-1 upon proteasome disruption by bortezomib. Actin is used as a loading control. (**c**) Induction of GFP::DDI-1 upon proteasome disruption by *rpt-5(RNAi)* is lost in *skn-1a* mutants. Scale bar 20 µm.

mutants (*Figure 4c*), suggesting that DDI-1 upregulation is mostly mediated by SKN-1A. The remaining weaker *ddi-1* upregulation in the *skn-1a* mutant may represent a second *skn-1a*-independent mechanism that couples DDI-1 levels to proteasome function.

## DDI-1 is required for proteolytic cleavage of SKN-1A downstream of ER trafficking

The EMS-induced *ddi-1* missense alleles that cause failure to activate *rpt-3::gfp* are clustered within the aspartic protease domain of DDI-1, and affect conserved residues that are thought to form the substrate-binding pocket of the enzyme (*Figure 5a,b*), suggesting that the protease activity of DDI-1 is required (*Sirkis et al., 2006*). We used CRISPR/Cas9 mutagenesis to generate a protease dead mutant containing two amino acid substitutions at conserved residues of the catalytic motif, including the aspartic acid residue that forms the active site (D261N, G263A). We additionally isolated a CRISPR-induced deletion that deletes most of the aspartic protease domain and introduces a frameshift, which we presume to be a null allele. Both mutations cause a similar, strong defect in *rpt-3::gfp* activation by the *pbs-5(mg502)* mutant, or upon proteasome RNAi, and cause a similar sensitivity to bortezomib (*Figure 5c*, data not shown).

*S. cervisiae* Ddi1 contains an N-terminal ubiquitin-like (UBL) domain and a C-terminal ubiquitin-associated (UBA) domain, but these domains are not detected by standard protein sequence comparisons with *C. elegans* DDI-1. To address the possibility that UBL or UBA domains with highly divergent sequence may be present in DDI-1, we generated N-terminally truncated (ΔN), and C-terminally truncated (ΔC) *gfp::ddi-1* transgenes. We tested their ability to rescue the bortezomib sensitivity phenotype of *ddi-1(mg571)* alongside wild-type *gfp::ddi-1* and an aspartic protease active site (D261N) mutant. The active site mutation abolished rescue by the *gfp::ddi-1* transgene, whereas the ΔN and ΔC truncated transgenes restored bortezomib sensitivity to near wild-type levels (*Figure 5—*

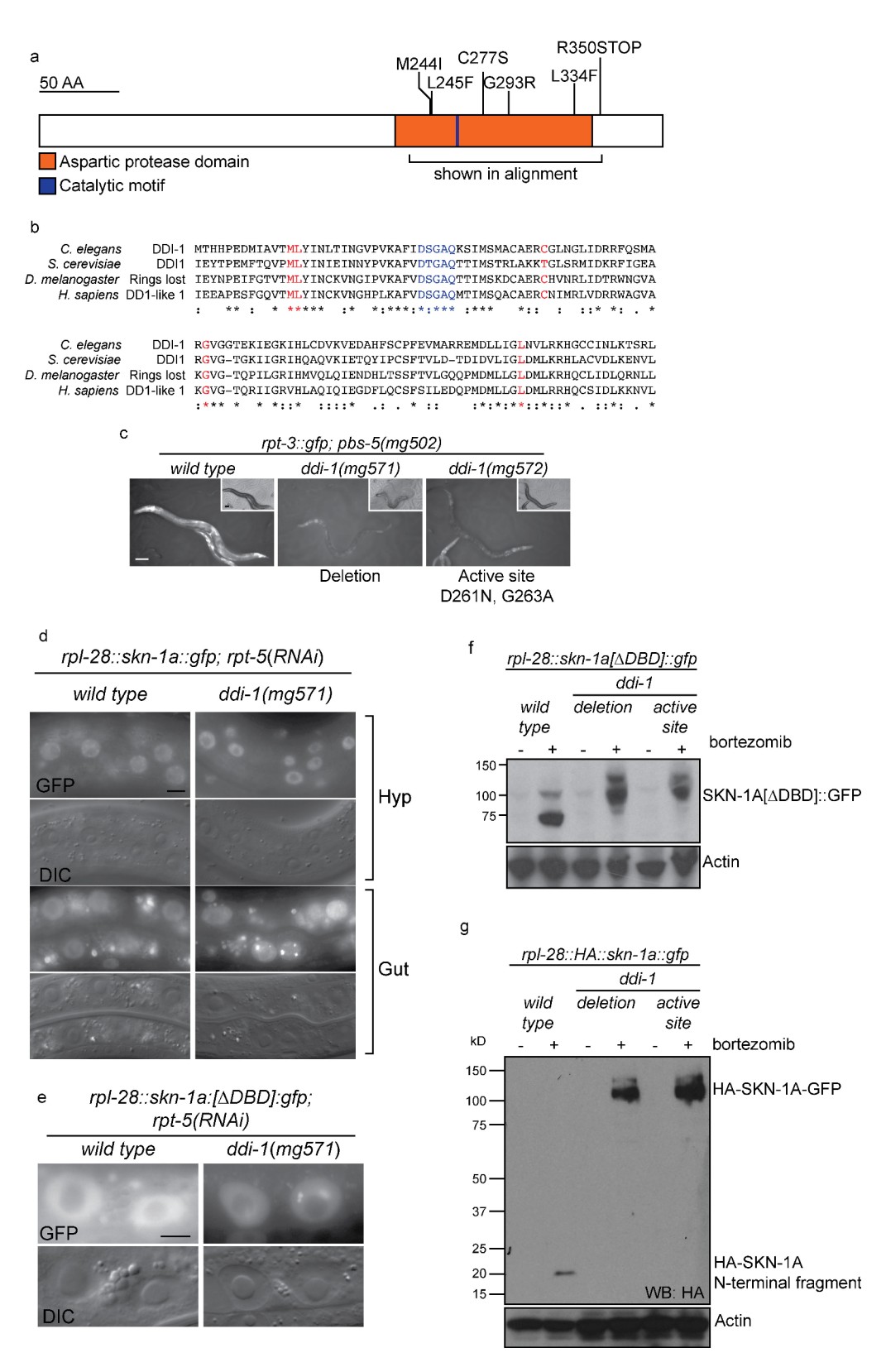

**Figure 5.** The DDI-1 aspartic protease is required for proteolytic activation of SKN-1A. (a) Schematic of the DDI-1 protein showing residues affected by EMS-induced loss of function alleles. (b) Multiple alignment of the aspartic protease domain of DDI-1. Red text indicates residues affected by mutations

*Figure 5 continued on next page*

*Figure 5 continued*

that disrupt *rpt-3::gfp* activation, blue text indicates the DS/TGAQ catalytic motif. (c) *rpt-3::gfp* induction in the *pbs-5(mg502)* mutant background (left panel), compared to animals carrying a *ddi-1* deletion (middle panel), or a point mutation affecting the catalytic motif (right panel). Scale bars 100 µm. (d) Localization of SKN-1A::GFP in wild-type and *ddi-1* mutant animals following disruption of proteasome function by *rpt-5(RNAi)*. SKN-1A::GFP nuclear localization is intact in the *ddi-1* mutant. In some gut cells of *ddi-1* mutant animals SKN-1A::GFP localizes to puncta that are not detected in the wild type. Scale bars 10 µm. (e) Localization of SKN-1A[ΔDBD]::GFP in gut nuclei of wild-type and *ddi-1* mutant animals. Nuclear foci of SKN-1A[ΔDBD]::GFP are found in *ddi-1* mutants, but not wild type. (f) Western blot showing expression and processing of SKN-1A[ΔDBD]::GFP in *ddi-1* mutant animals, treated with either solvent control (DMSO) or 5 ug/ml bortezomib, and blotted for GFP. In the *ddi-1* mutant animals, the major band detected is ~30 kD larger than in the wild type. (g) Western blot showing expression and processing of HA::SKN-1A:GFP in *ddi-1* mutants animals, treated with either solvent control (DMSO) or 5 ug/ml bortezomib, and blotted for HA. In the wild type, a ~20 kD band is detected in animals exposed to bortezomib. In *ddi-1* mutants this low molecular weight fragment is absent, and a ~110 kD band is detected. In (f) and (g) *ddi-1* mutations were *ddi-1 (mg571)*[deletion] or *ddi-1(mg572)*[active site] and actin is used as a loading control.

The following figure supplement is available for figure 5:

**Figure supplement 1.** The aspartic protease, but not the N- or C-terminal domains of DDI-1 are essential for resistance to bortezomib.

*figure supplement 1*). These data are consistent with the lack of conservation of the UBL and UBA domains of DDI-1, and confirm the essential role of DDI-1 aspartic protease activity.

In animals lacking DDI-1, SKN-1A::GFP localizes at normal levels to the nucleus upon proteasome disruption by *rpt-5(RNAi)*, suggesting that DDI-1 regulates SKN-1A function after nuclear localization of the transcription factor (*Figure 5d*). We noticed that SKN-1A::GFP occasionally showed abnormal localization within gut nuclei of *ddi-1* mutants, accumulating in highly fluorescent puncta. We observed this defect for both SKN-1A::GFP and SKN-1A[ΔDBD]::GFP, indicating that the DBD of SKN-1A is not required for this mis-localization (*Figure 5e*).

As in the wild type, SKN-1A[ΔDBD]::GFP does not accumulate in the absence of proteasome disruption in *ddi-1* mutants, indicating that the DDI-1 peptidase does not participate in constitutive degradation of SKN-1A by the proteasome (*Figure 5f*). SKN-1A[ΔDBD]::GFP accumulates to similar levels upon proteasome disruption by bortezomib in wild-type and *ddi-1* mutants, but in *ddi-1* mutants is ~20 kD larger than in the wild type, and approximates the expected size of SKN-1A [ΔDBD]::GFP. To test whether these differences reflect DDI-1-dependent proteolytic processing of SKN-1A, we generated a transgene that expresses full length SKN-1A with an N-terminal HA tag and a C-terminal GFP tag (HA::SKN-1A::GFP). The expression, localization and rescue activity of the dually tagged fusion protein is indistinguishable from that of the full length SKN-1A::GFP transgene. In wild-type animals carrying the HA::SKN-1A::GFP transgene, Western blotting for the HA tag reveals a ~20 kD band that accumulates specifically upon proteasome disruption by bortezomib treatment. In *ddi-1* deletion or active site mutants, a ~110 kD protein accumulates upon proteasome disruption, equivalent in size to full-length HA::SKN-1::GFP (*Figure 5g*). As such, SKN-1A is cleaved at a position approximately 20 kD from the N-terminus, and the protease active site of DDI-1 is required for this cleavage.

In *sel-1* and *sel-11* mutants, unprocessed HA::SKN-1A::GFP is present in both control and bortezomib treated animals, with some processed HA::SKN-1A::GFP appearing upon bortezomib treatment (*Figure 6a*). In *png-1* mutant animals HA::SKN-1A::GFP is incompletely processed following proteasome disruption (*Figure 6b*), suggesting that DDI-1 dependent cleavage of SKN-1A normally occurs after ER trafficking and deglycosylation. Further, in wild-type animals, an HA::SKN-1A::GFP transgene that lacks the putative transmembrane domain is not targeted for constitutive degradation, and is not subject to detectable proteolytic cleavage upon proteasome disruption (HA::SKN-1A [ΔTM]::GFP; *Figure 6c*). Although expressed, HA::SKN-1A[ΔTM]::GFP is unable to rescue the bortezomib sensitivity of *skn-1a* mutants (data not shown). These results indicate that ER trafficking is needed to target SKN-1A for later cleavage and activation by ddi-1.

## Discussion

Our genetic screen for mutants that fail to activate SKN-1 and dissection of the isoform-specific role of SKN-1A reveals the molecular details of proteasome surveillance. We show that SKN-1A is an ER

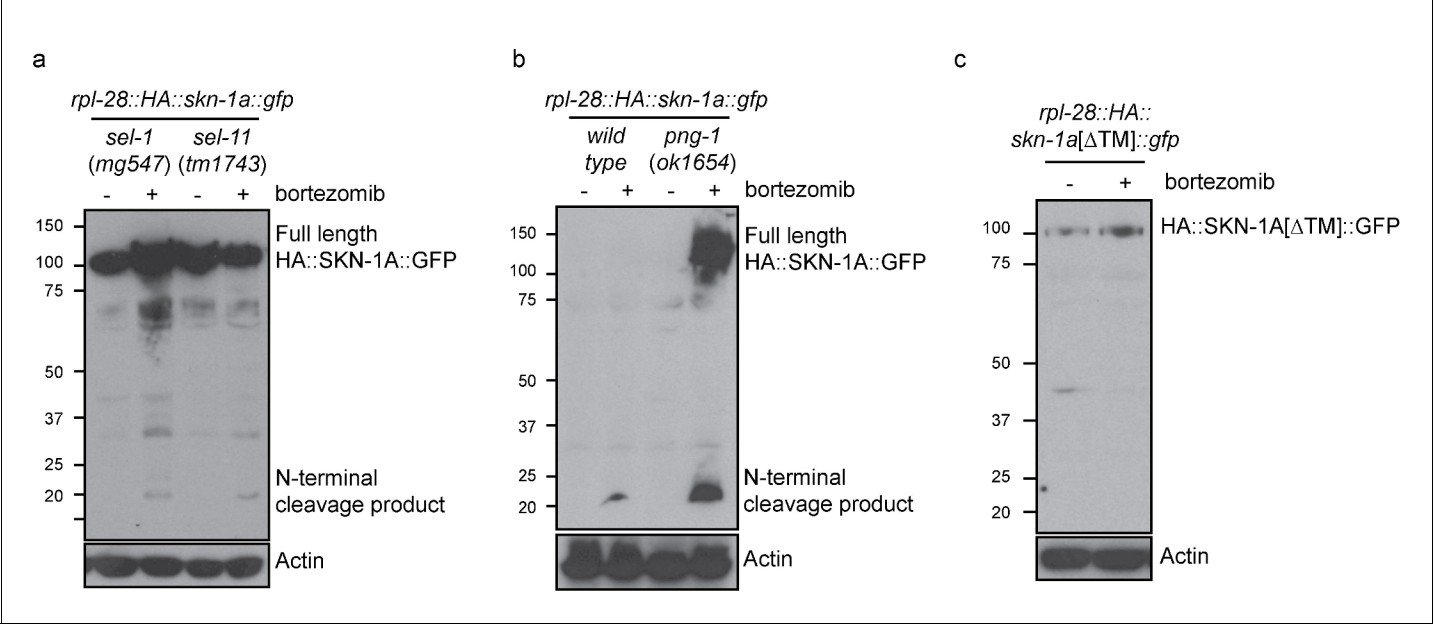

**Figure 6.** Proteolytic processing of SKN-1A occurs downstream of ER trafficking. (a) Western blot showing expression and processing of HA::SKN-1A:: GFP upon proteasome disruption by bortezomib in *sel-1* and *sel-11* mutant animals. ERAD-defective mutants accumulate similar levels of uncleaved (~110 kD) HA::SKN-1A::GFP in both absence and presence of bortezomib. A relatively small amount of the ~20 kD cleavage product accumulates only upon exposure to bortezomib. (b) Western blots comparing expression and processing of HA::SKN-1A::GFP upon proteasome disruption by bortezomib in wild-type and *png-1* mutant animals. Both cleaved (~20 kD) and uncleaved (smear ~100-150 kD) HA::SKN-1A::GFP accumulates upon bortezomib treatment in *png-1* mutants. (c) Western blot showing expression and processing of HA::SKN-1A[ΔTM]::GFP upon proteasome disruption by bortezomib, in otherwise wild-type animals. Uncleaved (~100 kD) HA::SKN-1A[ΔTM]::GFP is detected at similar levels under both conditions suggesting the protein is not subject to proteasomal degradation, and a low molecular weight cleavage product is not detected under either condition. For each experiment, mixed stage cultures were treated with either solvent control (-), or 5 ug/ml bortezomib (+) prior to collection for SDS-PAGE. HA:: SKN-1A::GFP/ HA::SKN-1A[ΔTM]::GFP is detected by anti-HA antibodies, Actin is used as a loading control.

The following figure supplement is available for figure 6:

**Figure supplement 1.** Model showing post-translational processing of SKN-1A by DDI-1.

associated protein that is normally targeted by the ERAD pathway for proteasomal degradation in the cytoplasm. Mutations affecting ERAD genes *sel-1*/SEL1/HRD3 and *sel-11*/HRD1, stabilize SKN-1A, but also disrupt localization of SKN-1A following proteasome disruption, due to a failure to efficiently release SKN-1A from the ER. Our data argues that following release from the ER, SKN-1A must be deglycosylated by PNG-1 and cleaved by DDI-1 to become fully active *Figure 6—figure supplement 1*.

The bortezomib sensitivity defect of *png-1* mutants is similar in strength to *skn-1a* mutants, and both *png-1* and *skn-1a* mutations are synthetic lethal with *rpn-10(RNAi)*. These similarities suggest that SKN-1A activity may be completely abolished in the absence of PNG-1; likely as a result of failure to deglycosylate SKN-1A following its retrotranslocation from the ER, although we cannot rule out the possibility that deglycosylation of other proteins contributes indirectly to SKN-1A function. Surprisingly, we were unable to generate *png-1; skn-1a* double mutants, apparently due to lethality of double mutant embryos (NL unpublished). This indicates that in *png-1* mutant animals, SKN-1A (likely in a glycosylated state) retains a function that is essential for development. This suggests that SKN-1A is not only important for proteasome homeostasis, but is also important for cellular homeostasis upon disruption of glycoprotein metabolism. Consistently, *skn-1* mutants are hypersensitive to tunicamycin, an inhibitor of protein glycosylation (*Glover-Cutter et al., 2013*).

NGLY1, the human PNG-1 orthologue, plays important roles in human development; NGLY1 deficiency, a recently described genetic disorder of protein deglycosylation, is caused by mutations at the NGLY1 locus (*Enns et al., 2014*). Failure to deglycosylate Nrf1, and consequent defects in

proteasome homeostasis, likely contributes to the symptoms associated with NGLY1 deficiency. As such our work identifies a pathway that may be targeted for treatment of NGLY1 deficiency - genetic screens for suppressors of the defective proteasome gene regulation of *png-1* mutants could indicate targets for drug development.

The expression of DDI-1 is dramatically responsive to proteasome inhibition, indicating that its synthesis or stability is coupled to surveillance of proteasome dysfunction. ChIP analysis of SKN-1 shows binding at the promoter element of the DDI-1 operon (*Niu et al., 2011*), suggesting that DDI-1 upregulation may occur via SKN-1 mediated transcriptional regulation, and we observed a defect in the upregulation of GFP::DDI-1 following proteasome disruption in *skn-1a* mutants. This suggests a positive feedback loop, wherein SKN-1A upregulates DDI-1, and DDI-1 promotes SKN-1A activation. Positive feedback may ensure a timely and robust response to proteasome inhibition.

Mutation of *ddi-1* causes defective regulation of *rpt-3::gfp* and increases sensitivity of *C. elegans* to proteasome inhibitors. SKN-1A is cleaved at a site ~20 kD from the N-terminus, and *ddi-1* is required for this cleavage. The requirement for its catalytic function strongly suggests that DDI-1 is the enzyme directly responsible for cleavage of SKN-1A, although we cannot rule out the possibility that DDI-1 acts upstream of an as-yet unidentified second protease. There are precedents for cascades of proteases, for example caspases in apoptosis, complement cascades in immunology, and thrombin cascades in blood clotting. Our genetic screen for proteasome surveillance defective mutants isolated six independent *ddi-1* alleles, but as yet no alleles of any other genes that encode proteases. This argues that either DDI-1 is the only protease in the pathway, or that any other proteases function redundantly or have other essential functions.

Uncleaved SKN-1A localizes to the nucleus in *ddi-1* mutants, so cleavage of SKN-1A is not essential for its nuclear localization, and SKN-1A cleavage may occur either in the nucleus or cytoplasm. Given that GFP::DDI-1 is largely (but not exclusively) nuclear under conditions of proteasome disruption, we speculate that DDI-1 cleavage of SKN-1A is nuclear. In either case, unusually for a membrane-associated transcription factor, SKN-1A is released from the ER by a mechanism that does not require proteolytic cleavage. DDI-1-dependent cleavage therefore activates SKN-1A by some other mechanism(s) downstream of ER release. Cleavage of SKN-1A may be required to remove domains in the N-terminus that interfere with its normal function in the nucleus. For example, retention of the hydrophobic transmembrane domain may be disruptive once the protein has been extracted from the ER membrane.

Mutant SKN-1A lacking the transmembrane domain is not subject to proteasomal degradation, and is not cleaved by DDI-1. So, in addition to serving to link SKN-1A levels to proteasome function via ERAD, ER trafficking of SKN-1A is important for subsequent DDI-1-dependent activation. The bortezomib sensitivity of *skn-1a* mutants (or *skn-1a* mutants carrying the transmembrane domain-lacking transgene) is more severe than that of *ddi-1* and ERAD mutants, so ER-association must also promote SKN-1A activation by additional mechanisms. As well as proteasome disruption, *skn-1* is implicated in responses to several endocrine and environmental stimuli (*Blackwell et al., 2015*). Modifications such as glycosylation that SKN-1A acquires in the ER may tailor its activity to respond to proteasome dysfunction, identifying these modifications and how they are regulated will be of interest.

*S. cerevisiae* Ddi1 contains an N-terminal UBL domain and a C-terminal UBA domain, This domain architecture is typical of extraproteasomal ubiquitin receptors, which play a role in recruiting ubiquitinated proteins to the proteasome (*Tomko and Hochstrasser, 2013*). *S. cerevisiae* Ddi1 binds to both ubiquitin and the proteasome, and participates in the degradation of some proteasome substrates (*Bertolaet et al., 2001*; *Gomez et al., 2011*; *Kaplun et al., 2005*; *Nowicka et al., 2015*), and synthetic genetic interactions with extraproteasomal ubiquitin receptor and proteasome subunit mutants supports a role for Ddi1 in proteasome function (*Costanzo et al., 2010*; *Díaz-Martínez et al., 2006*). Although the aspartic protease domain is highly conserved, DDI-1 in *C. elegans* and related nematodes apparently lacks both UBL and UBA domains, and the UBA domain is not found in mammalian Ddi1 orthologues, so it remains unclear whether Ddi1 orthologues function as extraproteasomal ubiquitin receptors in animals (*Nowicka et al., 2015*). The effect of *ddi-1* on development upon proteasome disruption by bortezomib is entirely dependent on *skn-1a*, indicating that DDI-1 promotes resistance to proteasome disruption via SKN-1A, rather than a general effect on proteasome function. Regardless, it will be of interest to determine whether DDI-1 binds to proteasomes and/or ubiquitin, and whether this affects its function in SKN-1A activation.

Activation of the mammalian SKN-1 homologue Nrf1 involves both deglycosylation and proteolytic cleavage, but the enzymes responsible are not known (*Radhakrishnan et al., 2014*; *Zhang et al., 2015*). However, a large scale screen for gene inactivations that render cells more sensitive to proteasome inhibitors supports a model that human DDI1 protease also processes Nrf1: DDI2 (one of two human orthologues of DDI-1) and Nrf1 were highly ranked hits in this screen that identified hundreds of gene inactivations that increase sensitivity of multiple myeloma cells to proteasome inhibitors (*Acosta-Alvear et al., 2015*). This suggests that DDI2 is required to cleave and activate Nrf1 in human cells. The site at which Nrf1 is cleaved during proteasome dysfunction has been identified (*Radhakrishnan et al., 2014*), but the primary sequence of this site is not conserved in SKN-1A. Comparisons of SKN-1A with its nematode orthologues reveals conservation at positions consistent with the ~20 kD cleavage product we have observed (NL unpublished). It is possible that DDI-1 and its substrate(s) have divergently co-evolved in different lineages. Thus, we suggest that DDI-1 and SKN-1A are core components of a conserved mechanism of proteasome surveillance in animals.

Here we have shown that correct post-translational processing of SKN-1A is essential for development if proteasome function is disrupted. Deregulated proteasome function a feature of aging and age-related disease (*Saez and Vilchez, 2014*; *Taylor and Dillin, 2011*). *skn-1* is a critical genetic regulator of longevity, and controls lifespan in part through regulation of proteasome function (*Blackwell et al., 2015*; *Steinbaugh et al., 2015*) As such, the SKN-1A processing pathway described here suggests the mechanism that links SKN-1/Nrf to proteasome function and longevity.

Proteasome inhibitors are important drugs in the treatment of multiple myeloma, but relapse and emergence of drug resistant tumors remains a challenge (*Dou and Zonder, 2014*). Nrf1 promotes survival of cancerous cells treated with proteasome inhibitors, and activation of this pathway might mediate resistance (*Acosta-Alvear et al., 2015*; *Radhakrishnan et al., 2010*; *Steffen et al., 2010*). Blocking the activation of proteasome subunit gene expression by Nrf1 has been proposed as a potential strategy to improve effectiveness of proteasome inhibitors in cancer treatment. The conserved SKN-1A/Nrf1 processing factors we have identified, particularly DDI-1, are ideal targets for such an approach.

## Materials and methods

### *C. elegans* maintenance and genetics

*C. elegans* were maintained on standard media at 20°C and fed *E. coli* OP50. A list of strains used in this study is provided in Supplementary *Table 1*. Mutagenesis was performed by treatment of L4 animals in 47 mM EMS for 4 hr at 20°C. RNAi was performed as described in *Kamath and Ahringer (2003)*. The *mgIs72[rpt-3::gfp]* integrated transgene was generated from *sEx15003* (*Hunt-Newbury et al., 2007*), using EMS mutagenesis to induce integration of the extrachromosomal array. Some strains were provided by the CGC, which is funded by NIH Office of Research Infrastructure Programs (P40 OD010440). *sel-11(tm1743)* was kindly provided by Shohei Mitani. *png-1 (ok1654)* was generated by the *C. elegans* Gene Knockout Project at the Oklahoma Medical Research Foundation, part of the International *C. elegans* Gene Knockout Consortium.

### Identification of EMS induced mutations by whole genome sequencing

Genomic DNA was prepared using the Gentra Puregene Tissue kit (Qiagen, #158689) according to the manufacturer's instructions. Genomic DNA libraries were prepared using the NEBNext genomic DNA library construction kit (New England Biololabs, #E6040), and sequenced on a Illumina Hiseq instrument. Deep sequencing reads were analyzed using Cloudmap (*Minevich et al., 2012*).

Following deep sequencing analysis, a number of criteria were taken into account to identify the causative alleles, as shown in *Supplemental file 1*. In many cases, the causative alleles were strongly suggested by the identification of multiple independent alleles for a given gene. Even for those genes only identified by a single allele, the strong functional connection with other independently mutated genes suggests that they are causative (e.g. isolation of multiple alleles of the *sel* gene class). We also obtained genetic linkage data supporting these assignments for most alleles. For most of the mutants considered, deep sequencing was performed using a DNA from a pool of 20–50 mutant F2s generated by outcrossing the original mutant strain to the parental (non-

mutagenised) background, which allowed us to use Cloudmap variant discovery mapping to identify the genetic linkage of the causative allele, or linkage was confirmed by testing linkage in crosses with strains carrying precisely mapped miniMos insertions. For *ddi-1* and *png-1*, we confirmed that disruption by an independent means (with an independently derived allele) has the same effect on *rpt-3::gfp* expression as the EMS-induced mutation.

## Identification of the *pbs-5(mg502)* mutation

The *mg502* allele was isolated in an EMS mutagenesis screen in which *mgIs72[rpt-3::gfp]* animals were screened for recessive mutations causing constitutive activation of GFP expression. The mutation was identified as described above. The *pbs-5(mg502)* lesion is a 122bp deletion in the promoter of CEOP1752. This operon consists of *K05C4.2* and *pbs-5*. Animals carrying this mutation show constitutive activation of *rpt-3::gfp*, but have normal growth and fertility under control conditions.

## Genome modification by CRISPR/Cas9

Guide RNAs were selected by searching the desired genomic interval for 'NNNNNNNNNNNNNNNNNNNRRNGG', using Ape DNA sequence editing software (http://biology-labs.utah.edu/jorgensen/wayned/ape/). All guide RNA constructs were generated by Q5 site directed mutagenesis as described (*Dickinson et al., 2013*). Repair template oligos were designed as described (*Paix et al., 2014*; *Ward, 2015*). Injections were performed using the editing of *pha-1* (to restore *e2123ts*) or *dpy-10* (to generate *cn64* rollers) as phenotypic co-CRISPR markers (*Arribere et al., 2014*; *Ward, 2015*). Injection mixes contained 60 ng/ul each of the co-CRISPR and gene of interest targeting Guide RNA/Cas9 construct, and 50 ng/ul each of the co-CRISPR and gene of interest repair oligos. Guide RNA and homologous repair template sequences are listed in *Supplemental file 1*.

## Transgenesis

Cloning was performed by isothermal/Gibson assembly (*Gibson et al., 2009*). All plasmids used for transgenesis are listed in *Supplemental file 1*. All miniMos constructs were assembled in pNL43, a modified version of pCFJ909 containing the pBluescript MCS, and are described in more detail below. MiniMos transgenic animals were isolated as described, using *unc-119* rescue to select transformants (*Frøkjaer-Jensen et al., 2014*).

## MiniMos constructs

### *skn-1* constructs

To generate constructs that express specific SKN-1 isoforms, we cloned *skn-1c* and *skn-1a* 'minigenes'. The *skn-1c* minigene was assembled from 3 genomic fragments from the *skn-1* locus: (1) *skn-1c* exon 1, (2) *skn-1c* exons 2 & 3, (3) *skn-1c* exons 4, 5 & 6. These three fragments were fused in frame to generate a *skn-1c* coding sequence lacking the two largest introns that may contain regulatory information. The *skn-1a* minigene was assembled by fusing two additional fragments, corresponding to the first two exons of *skn-1a*, to the 5' end of the *skn-1c* minigene, eliminating the large intron that contains the presumed *skn-1c* promoter. These minigenes were then fused in frame with an N-terminal HA tag and/or a C-terminal GFP tag, and inserted into pNL43 with the *rpl-28* promoter (605 bp immediately upstream of the *rpl-28* start codon) and *tbb-2* 3'UTR (376 bp immediately downstream of the *tbb-2* stop codon). *skn-1a[ΔDBD]::gfp* and *skn-1c[ΔDBD]::gfp* were generated by deleting sequence corresponding to the final 94 amino acids from the each construct (i.e. amino acids 530–623 of SKN-1A and amino acids 440–533 of SKN-1C) by isothermal assembly using appropriate fragments of the respective minigenes. *skn-1a[ΔTM]::gfp* was generated by deleting sequence corresponding to amino acids 39–59 of SKN-1A, by isothermal assembly of appropriate fragments of the *skn-1a* minigene.

### *ddi-1* constructs

The genomic *C01G5.6/ddi-1* coding sequence was fused in frame with GFP at the N-terminus. The *gfp::ddi-1* fragment was inserted into pNL43 with the *ddi-1* promoter (803 bp immediately upstream of the start codon), and the *tbb-2* 3'UTR. The *gfp::ddi-1*[D261N] construct was generated by site-

directed mutagenesis, and the N- and C-terminal truncation constructs were generated by isothermal assembly using appropriate fragments of the *ddi-1* genomic coding sequence.

## Microscopy

Low magnification bright field and GFP fluorescence images (those showing larval growth and *rpt-3:: gfp* expression) were collected using a Zeiss AxioZoom V16, equipped with a Hammamatsu Orca flash 4.0 digital camera camera, and using Axiovision software. High magnification differential interference contrast (DIC) and GFP fluorescence images (those showing SKN-1A::GFP and GFP::DDI-1 expression) were collected using a Zeiss Axio Image Z1 microscope, equipped with a Zeiss AxioCam HRc digital camera, and using Axiovision software. Images were processed using ImageJ software. For all fluorescence images, any images shown within the same figure panel were collected together using the same exposure time and then processed identically in ImageJ.

## Bortezomib sensitivity assays

Bortezomib sensitivity was assessed by the ability of L1 animals to develop in the presence of a range of concentrations of bortezomib (LC Laboratories, #B1408). The assays were carried out in liquid culture in ½ area 96 well plates (Corning, #3882). Each well contained a total volume of 35uL. We mixed ~15 L1 larvae with concentrated *E. coli* OP50 suspended in S-basal (equivalent to bacteria from ~200 uL of saturated LB culture), supplemented with 50 ug/ml Carbenicillin, and the desired concentration of bortezomib. All treatment conditions contained 0.01% DMSO. The plates were sealed with Breathe-Easy adhesive film (Diversified Biotech, #9123–6100). The liquid cultures were incubated for 4 days at 20°C and then *C. elegans* growth was manually scored under a dissecting microscope. Growth was scored into three categories: (1) Normal - indistinguishable from wild type grown in DMSO control, most animals reached adulthood; (2) Delayed development - most animals are L3 or L4 larvae; (3) Larval arrest/lethal - all animals are L1 or L2 larvae. For each genotype all conditions were tested in at least 2 replicate experiments.

## Western blot following bortezomib treatment

Drug treatments were performed in liquid culture in 6-well tissue culture plates. In each well we mixed *C. elegans* suspended in S-basal (~1000–2000 worms collected from a mixed stage culture grown at 20°C on NGM agar plates) and *E. coli* OP50 in S-Basal (equivalent to *E. coli* from ~4 mL saturated LB culture), supplemented with 50 ug/ml Carbenicillin, and the desired concentration of bortezomib, and made the mixture up to a final volume of 700 ul. All wells contained 0.01% DMSO. The tissue culture plates were sealed with Breathe-Easy adhesive film and incubated at 20°C for 7–9 hr. After the treatment the animals were collected to 1.5 ml microcentrifuge tubes, washed twice in PBS to remove bacteria and the worm pellet was snap frozen in liquid nitrogen and stored at −80°C. The worm pellet was briefly thawed on ice, mixed with an equal volume of 2x Sample buffer (20% glycerol, 120 mM Tris pH 6.8, 4% SDS, 0.1 mg/ml bromophenol blue, 5% beta mercaptoethanol), heated to 95°C for 10 min, and centrifuged at 16,000g for 10 min to pellet debris. SDS-PAGE and western blotting was performed using NuPAGE apparatus, 4–12% polyacrylamide Bis-Tris pre-cast gels (Invitrogen, #NP0321) and nitrocellulose membranes (Invitrogen, #LC2000) according to the manufacturer's instructions. The following antibodies were used: mouse anti-GFP (Roche; #11814460001); HRP-conjugated mouse anti-HA (Roche, # 12013819001), mouse anti-Actin (Abcam; #3280).

## Multiple alignment of protein sequences

Multiple alignment was performed using Clustal Omega (www.ebi.ac.uk/tools/clustalo).

## Acknowledgements

We would like to thank Ulandt Kim for high-throughput sequencing. We thank Fei Ji and Ruslan Sadreyev for bioinformatics advice. We thank Fred Goldberg and Keith Blackwell and members of their labs for useful discussions. This work was supported by an NIH R01 grant to GR [R01 AG016636]. NL was supported by a Human Frontier Science Program Postdoctoral Fellowship.

## Additional information

### Funding

| Funder | Grant reference number | Author |
| --- | --- | --- |
| National Institutes of Health | R01 AG016636 | Gary Ruvkun |
| Human Frontier Science Program | Long Term Fellowship | Nicolas J Lehrbach |

The funders had no role in study design, data collection and interpretation, or the decision to submit the work for publication.

### Author contributions

NJL, Conception and design, Acquisition of data, Analysis and interpretation of data, Drafting or revising the article; GR, Conception and design, Analysis and interpretation of data, Drafting or revising the article

### Author ORCIDs

Gary Ruvkun, http://orcid.org/0000-0002-7473-8484

## Additional files

### Supplementary files

• Supplementary file 1. Supplementary Tables 1-4.xlsx. Contains 4 Supplementary tables: Table 1. *C. elegans* strains used in this study. Table 2. CRISPR guide RNA constructs and repair template sequences. Table 3. Constructs used to generate miniMos transgenics. Table 4. Evidence used in identification of EMS-induced mutations that cause failure to activate *rpt-3::gfp* expression.

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
