## [Decision Letter]

Thank you for submitting your article "Proteasome dysfunction triggers activation of SKN-1A/Nrf1 by the aspartic protease DDI-1" for consideration by *eLife*. Your article has been reviewed by three peer reviewers, one of whom, Andrew Dillin, is a member of our Board of Reviewing Editors, and the evaluation has been overseen by Kevin Struhl as the Senior Editor. The following individuals involved in review of your submission have agreed to reveal their identity: Andrew Dillin (Reviewing Editor and Reviewer #1) and Raymond J Deshaies (Reviewer #2).

The reviewers have discussed the reviews with one another and the Reviewing Editor has drafted this decision to help you prepare a revised submission. As you find within the reviews, each reviewer was very enthusiastic about this body of work. While we each had minor comments and questions, we believe the body of work is a significant advance for the field. We ask that you provide a response to each comment from the reviewers.

Reviewer #1:

In this body of work Ruvkun and colleagues begin to query proteasome regulation using elegant genetics and the nematode *C. elegans*. In this data packed manuscript they outline the role of DDI-1, an aspartic protease, that moves to the nucleus under conditions of proteasome stress. The protease domain is more than likely required to regulate a single isoform of *skn-1*, thereby regulating its nuclear activities. Throughout there is great genetic interaction studies between *ddi-1, skn-1* and components of the ERAD pathway (which are required to liberate *skn-1* from the ER). There is a cornucopia of findings throughout and almost every stone has been turned.

Minor comments:

The western bot shown in Figure 4 is blown out. It’s ok to show it, but the authors should also include a lighter exposed one.

It would also help the reader to include a model describing *skn-1* and *ddi-1* under static vs. proteasome stress.

The induction of rapt is followed by GFP reporters. Does the endogenous gene also show similar regulation?

Lastly, the movement of *ddi-1* to the nucleus is shown during proteasome stress. Do other stresses also regulate its movement?

Reviewer #2:

In this manuscript, Lehrbach and Ruvkun report the identification of *C. elegans ddi-1* as a gene that is required for the activation of proteasome gene transcription in response to chemical inhibition of the proteasome or knockdown of proteasome subunits. It was shown over 10 years ago that inhibition of the proteasome in mammalian cells results in transcriptional induction of genes that encode proteasome subunits. Six years ago, the Deshaies group showed that this induction requires Nrf1 in mouse cells, and this was followed by a report from the Kruger group that the Nrf1 ortholog TCF11 mediates the response in human cells. They also showed that TCF11 is a substrate for the ERAD pathway. More recently, the Deshaies group reported the surprising observation that most of Nrf1 enters the ER lumen during its biogenesis, but is quickly retrotranslocated to the cytosol in p97-dependent manner, where it is clipped on the C-terminal side of tryptophan-103. This retrotranslocation and clipping were shown to be essential for Nrf1 activation. However, the protease was not identified, although a subsequent report from Fred Goldberg's group claimed that Nrf1 processing is carried out by partially inhibited proteasomes.

Lehrbach and Ruvkun carried out a screen to identify genes required for activation of the *C. elegans* ortholog of Nrf1, SKN-1A. They identify DDI-1, which is an aspartyl protease of largely unknown function, that in budding yeast doubles as a substrate receptor for the proteasome. They also identify ERAD genes including PNG-1, which is the *C. elegans* ortholog of the protein N-glycosidase enzyme NGLY1 that cleaves asparagine-linked oligosaccharides from retrotranslocated proteins. Overall, this is a compelling manuscript that makes an important finding that is likely to have broad ramifications not only for Nrf1 and protein homeostasis, but also for aging, neurodegeneration, and possibly cancer therapy. In my view, this manuscript should be an exceptionally high priority for *eLife*.

I have several comments, which are all modest in nature:

1) At the outset of Results, it should be explained what *pbs-5(mg502*) is, and what is the rationale for its use. I had to do a google search to figure out it is a mutation in a 20S proteasome subunit.

2) What happens in a *png-1* ERAD double mutant in terms of growth in bortezomib? Is the enhanced proteasome sensitivity of png-1 compared to ERAD mutants due to accumulation of glycosylated ERAD substrates in cytosol, or are the ERAD mutants simply leaker? i.e. which phenotype is epistatic?

3) What happens in the intestine in *skn1-a*? It is stated in the manuscript that SKN-1C is expressed in the intestine. Does this provide SKN-1A function in the intestine?

4) The authors make the comment, "We generated similar transgenes to express tagged full length and truncated SKN-1C, but did not observe any effect of proteasome disruption (data not shown)." What exactly is meant by this? Was SKN-1C not expressed under either condition (-/+ proteasome inhibition), or was SKN-1C equally expressed under both conditions?

5) Can the authors please comment on whether the cleavage site reported by Radhakrishnan et al. is in a region that is conserved in SKN-1A? How does the cleavage site in the human protein relate to the N-terminus of the ∆TMD construct?

6) Some of the references are not complete. This includes Arribere, Ayyadevara, Blackwell, Dickinson, Paix, Radhakrishnan 2014 (why is M.S. Brown an author here?), Sha, Zhang 2013.

Reviewer #3:

The transcription factor SKN-1/NRF1 is best known for its roles in oxidative and xenobiotic stress response pathways, but in recent years has become increasingly implicated in protein homeostasis. It responds to translational inhibition, upregulating the stress response, it controls proteasomal gene expression, upregulating target genes in response to proteasomal compromise. Finally it plays an important role in the ER stress response and autophagy regulation. However, the pathways governing these regulatory networks and their potential interconnections are not well understood.

In this work, Ruvkun and co-workers perform genetic screens in *C. elegans* for loci required for the SKN-1 regulatory response to proteasome inhibition. Interestingly, they identify a number of genes involved in ER associated degradation (*sel-1, sel-11, sel-9, png-1*), as well as the aspartyl protease ddi-1. They further show that an ER localized isoform of SKN-1a/NRF-1 translocates to the nucleus upon proteasome knockdown, where it upregulates its targets. ERAD knockdown also facilitate SKN-1a stabilization, although without nuclear localization, suggesting that ERAD components control SKN-1 availability from the ER.

The authors focus predominately on characterizing DDI-1, a conserved aspartyl protease whose function is poorly understood. They find that DDI-1 is induced upon proteasomal dysfunction in a SKN-1 dependent manner, and localizes largely to the nucleus. They find that DDI-1 plays a role in proteolytic processing of SKN-1A, which is likely important for its activity.

This is an excellent study that uses the power of genetics to unravel a very topical area, how proteasomal status regulates NRF activity and proteasome surveillance. This work gives key insight into how this might occur via ERAD and proteolytic regulation, providing a plausible link of NRF1 activation with the ER stress response and ERAD. The authors focus on the role of DDI-1 predominately because it is not well studied, thereby lending new insight. Among other things, this study raises the question of how mechanistically ERAD mobilizes SKN-1A from the ER, but perhaps this would be the subject of a whole other study.

I only have a few concerns/questions:

Major: The major question is to what extent does proteasome inhibition link to canonical ERAD signaling?

1) Do other ERAD components, such as CDC-48, play any role in SKN-1 dependent proteasomal activation?

2) Does proteasome compromise trigger the ER stress response, (and thereby SKN-1a activation? i.e. does proteasome knockdown induce UPR or ERAD?

3) Does ddi-1 loss block all transcriptional SKN-1a activity or only targets linked to ERAD and proteasome perturbation?

4) Does ddi-1 play any role in ERAD? Is it required to degrade an ERAD substrate?

---

## [Author Response]

Reviewer #1:

In this body of work Ruvkun and colleagues begin to query proteasome regulation using elegant genetics and the nematode C. elegans. In this data packed manuscript they outline the role of DDI-1, an aspartic protease, that moves to the nucleus under conditions of proteasome stress. The protease domain is more than likely required to regulate a single isoform of skn-1, thereby regulating its nuclear activities. Throughout there is great genetic interaction studies between ddi-1, son-1 and components of the ERAD pathway (which are required to liberate skn-1 from the ER). There is a cornucopia of findings throughout and almost every stone has been turned.

*Minor comments:*

*The western bot shown in Figure 4 is blown out. It’s ok to show it, but the authors should also include a lighter exposed one.*

We have added a shorter exposure of the same blot to Figure 4.

It would also help the reader to include a model describing skn-1 and ddi-1 under static vs. proteasome stress.

We have added a model as a figure supplement to the final figure.

The induction of rapt is followed by GFP reporters. Does the endogenous gene also show similar regulation?

Endogenous *rpt-3* has been shown by the Blackwell lab to be induced in a *skn-1*-dependent manner in response to proteasome disruption (Li et al, 2011). This was one of the reasons we chose to use the *rpt-3::gfp* reporter. We have added a reference to this point in the results section.

Lastly, the movement of ddi-1 to the nucleus is shown during proteasome stress. Do other stresses also regulate its movement?

This is an interesting question – but we think addressing this properly requires a separate study. We are planning to use the reagents we have generated (DDI-1 and SKN-1A fusion proteins) to perform a more comprehensive survey of the stresses that stimulate DDI-1 to enter the nucleus and cleave SKN-1A.

Reviewer #2:

*[…] I have several comments, which are all modest in nature:*

*1) At the outset of Results, it should be explained what pbs-5(mg502) is, and what is the rationale for its use. I had to do a google search to figure out it is a mutation in a 20S proteasome subunit.*

We have clarified this section of the results. We now point out that *pbs-5* encodes the beta 5 20S proteasome subunit of *C. elegans*. And we now indicate that we used this allele for our screen because it causes *skn-1* dependent activation of *rpt-3::gfp*, but otherwise does not affect viability. (Aside – *pbs-5(mg502)* is one of ~20 proteasome subunit mutations we isolated in genetic screens for constitutive activation of *rpt-3::gfp*, The other proteasome mutations cause lethality when *skn-1* is inactivated by RNAi. Using any of these other mutant strains would have made our screen technically much more challenging, since failure to activate *skn-1* would cause lethality).

*2) What happens in a png-1 ERAD double mutant in terms of growth in bortezomib? Is the enhanced proteasome sensitivity of png-1 compared to ERAD mutants due to accumulation of glycosylated ERAD substrates in cytosol, or are the ERAD mutants simply leaker? i.e. which phenotype is epistatic?*

This is an important question, but we think one best left as part of a future study that more fully addresses the role of ERAD and PNG-1 in SKN-1A activation. (Aside – we have only done the experiment once, and replicates will be needed to confirm, but it appears that the *png-1* phenotype is epistatic in *png-1; sel-1* double mutants. i.e. the sensitivity of the *png-1* mutant is probably not due to accumulation of glycosylated ERAD substrates in the cytoplasm.)

*3) What happens in the intestine in skn1-a? It is stated in the manuscript that SKN-1C is expressed in the intestine. Does this provide SKN-1A function in the intestine?*

Our experiments with *rpt-3::gfp* suggest that in the context of proteasome dysfunction caused by RNAi or bortezomib treatment, SKN-1B and SKN-1C do not provide function. In *skn-1a* mutant animals we have not observed activation of *rpt-3::gfp* in any tissue, including the intestine, where skn-1C is expressed. We have added a sentence to the results to emphasize this point. One interpretation of our data may be that SKN-1A and SKN-1C (even though they have the same DNA binding domain) are regulated by distinct mechanisms. Indeed, as SKN-1A is the only SKN-1 isoform with a transmembrane domain it is unlikely that ERAD and DDI-1 play a role in SKN-1C activation. This may be analogous to the different mechanisms regulating Nrf1 and Nrf2 in mammals. A full answer to this question will require a deeper understanding of the shared and distinct molecular mechanisms that govern activation of SKN-1A and SKN-1C.

*4) The authors make the comment, "We generated similar transgenes to express tagged full length and truncated SKN-1C, but did not observe any effect of proteasome disruption (data not shown)." What exactly is meant by this? Was SKN-1C not expressed under either condition (-/+ proteasome inhibition), or was SKN-1C equally expressed under both conditions?*

SKN-1C::GFP expression was not detectable under either condition. We admit that this negative result in itself is somewhat weak, but we think it is important when taken together with the genetic requirement/sufficiency of *skn-1a*. We favor keeping this sentence in the manuscript – this result (and the existence of the *skn-1c::gfp* transgenes) should be of interest to SKN-1 aficionados.

*5) Can the authors please comment on whether the cleavage site reported by Radhakrishnan et al. is in a region that is conserved in SKN-1A? How does the cleavage site in the human protein relate to the N-terminus of the ∆TMD construct?*

The sequence conservation between SKN-1A and Nrf1 is restricted to the DNA binding domain near the C-terminus of the protein; the primary sequence of the cleavage site (AAs 99-108 of Nrf1) does not appear to be conserved. Even within nematodes, the N-terminus of SKN-1A is not highly conserved – although there are two relatively well conserved stretches at AA 122-135 and AA 155-167.

The size (~20kD) of the N-terminal HA::SKN-1A fragment that is present in wild type animals treated with bortezomib, suggests that the cleavage site is somewhere in the vicinity of AAs 140-180 of SKN-1A. This putative cleavage site is some distance from the TMD at AAs 39-59, so we think that the failure to cleave ∆TMD SKN-1A is not a consequence of disrupting the cleavage site. We are working to map the cleavage site using HA::SKN-1A::GFP transgenes with small deletions within AAs 130-180. Future analysis of cleavage-resistant SKN-1A mutants will be needed to clarify the relationship between ER-trafficking and cleavage, and whether there are conserved features between the Nrf1 and SKN-1A cleavage sites.

We have added sentences to the discussion to point out the lack of conservation of the Nrf1 cleavage site and the presence of potential conserved SKN-1A cleavage site(s) in nematode Nrf1/SKN-1A orthologues.

*6) Some of the references are not complete. This includes Arribere, Ayyadevara, Blackwell, Dickinson, Paix, Radhakrishnan 2014 (why is M.S. Brown an author here?), Sha, Zhang 2013.*

We have corrected the mistakes in the references.

*Reviewer #3:*

*[…] I only have a few concerns/questions:*

*Major: The major question is to what extent does proteasome inhibition link to canonical ERAD signaling?*

This is an important question, but considerable work will be required to tease out the interaction between SKN-1A, ERAD, the UPR and possibly other stress response pathways. We agree with the reviewer’s suggestion above that this would be best addressed by a separate study.

*1) Do other ERAD components, such as CDC-48, play any role in SKN-1 dependent proteasomal activation?*

Yes, but it’s complicated. *C. elegans* has two near-identical genes encoding CDC-48 (*cdc-48.1* and *cdc-48.2*) which appear to be redundant – this might explain why no *cdc-48* alleles were isolated in our mutagenesis screen. RNAi of either gene apparently silences both paralogs and is lethal. *cdc-48.1/2* RNAi of wild type animals causes mild activation of *rpt-3::gfp*, and *cdc-48* RNAi of *pbs-5(nic67)* mutants causes a mild reduction in *rpt-3::gfp* expression. RNAi-mediated silencing of the *cdc-48* cofactors *ufd-1* or *npl-4* each cause activation of *rpt-3::gfp* in wild type animals. Overall these data suggest that the CDC-48/UFD-1/NPL-4 complex may play a role as both a positive and negative regulator of SKN-1A, and more work will be needed to clarify its role.

*2) Does proteasome compromise trigger the ER stress response, (and thereby SKN-1a activation? i.e. does proteasome knockdown induce UPR or ERAD?*

There is evidence that the UPR is activated by proteasome dysfunction in *C. elegans*; e.g. Proteasome RNAi induces expression of *hsp-4::gfp*, a UPR reporter that is regulated by XBP-1. Since the UPR is thought to regulate ERAD, it is reasonable to suspect that the UPR could influence, at least under some circumstances, the activation of SKN-1A. We think testing this is beyond the scope of this study but would be worth doing in future.

*3) Does ddi-1 loss block all transcriptional SKN-1a activity or only targets linked to ERAD and proteasome perturbation?*

We do not know yet. We plan to perform RNAseq and SKN-1A ChIP experiments to examine this.

*4) Does ddi-1 play any role in ERAD? Is it required to degrade an ERAD substrate?*

Constitutive degradation of SKN-1A requires the ERAD factors SEL-1 and SEL-11, but does not require DDI-1, so this is at least one ERAD substrate that is still degraded in *ddi-1* mutants. Relatively little is known about endogenous ERAD substrates in *C. elegans*, so time-consuming experiments to find and characterize such ERAD substrates would be required to address this question further.

Related to both this point and point (1) above, it would be interesting to determine which (if any) extraproteasomal ubiquitin receptors participate in ERAD in *C. elegans*, and whether these affect SKN-1A degradation or activation. Again, this is best the focus for a future study.